# Antioxidant Activity of *Acanthopanax senticosus* Flavonoids in H_2_O_2_-Induced RAW 264.7 Cells and DSS-Induced Colitis in Mice

**DOI:** 10.3390/molecules27092872

**Published:** 2022-04-30

**Authors:** Jianqing Su, Xinyu Zhang, Qibin Kan, Xiuling Chu

**Affiliations:** College of Agronomy and Agricultural Engineering, Liaocheng University, Liaocheng 252000, China; xinxinxinxinyuyuyu@163.com (X.Z.); kanqibin@163.com (Q.K.)

**Keywords:** oxidative stress, in vitro and in vivo, *Acanthopanax senticosus*, flavonoids

## Abstract

The redox reaction is a normal process of biological metabolism in the body that leads to the production of free radicals. Under conditions such as pathogenic infection, stress, and drug exposure, free radicals can exceed normal levels, causing protein denaturation, DNA damage, and the oxidation of the cell membrane, which, in turn, causes inflammation. *Acanthopanax senticosus (A. senticosus)* flavonoids are the main bioactive ingredients with antioxidant function. H_2_O_2_-treated RAW 264.7 cells and DSS-induced colitis in mice were used to evaluate the antioxidant properties of *A. senticosus* flavonoids. The results show that *A. senticosus* flavonoids can significantly downregulate the levels of ROS and MDA in H_2_O_2_-treated RAW 264.7 cells and increase the levels of CAT, SOD, and GPx. *A. senticosus* flavonoids can also increase the body weights of DSS-induced colitis mice, increase the DAI index, and ameliorate the shortening of the colon. ELISA experiments confirmed that *A. senticosus* flavonoids could reduce the level of MDA in the mouse serum and increase the levels of SOD, CAT, and GPx. Histopathology showed that the tissue pathological changes in the *A. senticosus* flavonoid group were significantly lower than those in the DSS group. The Western blot experiments showed that the antioxidant capacity of *A. senticosus* flavonoids was accomplished through the Nrf2 pathway. In conclusion, *A. senticosus* flavonoids can relieve oxidative stress in vivo and in vitro and protect cells or tissues from oxidative damage.

## 1. Introduction

Reactive oxygen species (ROS) are formed during an organism’s metabolism and include superoxide anions, hydrogen peroxide, and hydroxyl radicals, as well as free radicals. The body can modulate the ROS level with antioxidant enzymes and non-enzymatic scavengers. If the levels of oxidants and antioxidants become unbalanced, free radicals can cause the oxidation of lipids, proteins, and DNA [1,2], causing damage at a molecular level that may lead to disease [3,4]. The use of antioxidants is an important means of inhibiting oxidative stress, but the long-term use of chemically synthesized antioxidants, such as dibutyl hydroxytoluene (BHT) and tert-butyl hydroquinone (TBHQ), is considered harmful to the body [5]. Therefore, the research of natural antioxidants is another solution. Flavonoids are a large class of secondary metabolites of plant polyphenols, [6] which are produced by plants, and they have significant antioxidant properties. Flavonoids are important biologically active components of *A. senticosus* [7] that have significant effects in inhibiting inflammation, immune regulation, lowering blood sugar, hindering the proliferation of cancer cells, and resisting internal and external stress [8,9,10,11,12]. *A. senticosus* flavones have a better radical-scavenging ability than syringin and eleutheroside E [13,14]. The antioxidant effects of extracts of different parts of *A. senticosus* were evaluated in vitro, and the results showed that the extract of *A. senticosus* fruit have the strongest antioxidant effects according to ABTS and FRAP and have the most reducing power and ORAC; the extract of *A. senticosus* leaves have the strongest effects according to DPPH radical scavenging ability [15]. The correlation between the bioactive components of different parts of *A. senticosus* and antioxidant activities was investigated in H_2_O_2_-treated PC12 cells, and it was found that the roots of *A. senticosus* contained more antioxidant components than the seeds and leaves [16].

Macrophages are important immune cells that phagocytose pathogenic microorganisms [17]. In order to phagocytically remove pathogenic microorganisms, macrophages produce reactive oxygen species. However, some bacteria [18] and viruses [19] can promote the production of reactive oxygen species in macrophages to such high levels that it leads to the apoptosis of the macrophages [20]. Thus, macrophages are often used as models of oxidative stress in drug antioxidant tests, and hydrogen peroxide is usually used as a stimulus source.

The digestive tract is especially prone to oxidative stress due to the invasion of pathogenic microorganisms and toxins and heavy metals from food. Inflammatory bowel disease (IBD) is a series of chronic non-specific intestinal inflammatory diseases; ulcerative colitis (UC) [21] is one of these diseases, and the course of UC is chronic and recurrent [22]. Studies have found that reactive oxygen free radicals are involved in regulating the development of nonspecific intestinal inflammatory disease, including colitis and inflammatory bowel disease [23,24]. DSS-induced colitis is usually used as a model to study the oxidative stress induced by drugs.

However, there are relatively few systematic in vitro and in vivo studies on the antioxidant activities of flavonoids from *A. senticosus*. In this study, the antioxidant activities of flavonoids from *A. senticosus* were investigated in vitro and in vivo. The antioxidant mechanisms were studied through in H_2_O_2_-RAW264.7 cells and DSS-mice models.

## 2. Results

### 2.1. In Vitro Experiment

#### 2.1.1. Cytotoxicity Test

In order to determine the impact of *A. senticosus* flavonoids on the survival rate of RAW 264.7 cells and obtain the test drug concentration, the cells were treated with graded doses of *A. senticosus* flavonoids for 24 h. Afterward, the cell viability was determined using the CCK-8 kit. As shown in Figure 1A, after RAW 264.7 mouse macrophages treated with different concentrations of *A. senticosus* flavonoids for 24 h, it was found that *A. senticosus* flavonoids in the concentration range of 15–120 mg/L promoted the proliferation of macrophages in a dose-dependent manner. When the *A. senticosus* flavonoid concentration reached 150 mg/L, the viability of RAW 264.7 cells decreased compared with the blank group (*p* < 0.05). Therefore, the safe concentrations of *A. senticosus* flavonoids are 15~120 mg/L. Under this concentration range, *A. senticosus* flavonoids have a proliferative effect on RAW 264.7 cells. If the concentration of *A. senticosus* flavonoids is over 120 mg/L, they have a toxic effect on the cells, causing cell damage and cell death. Thus, concentrations of 30, 60, and 120 mg/L were used in the next assay.

#### 2.1.2. H_2_O_2_-Induced Oxidative Stress in RAW 264.7 Cells

In order to determine the test concentration of H_2_O_2_ in RAW 264.7 cells, RAW 264.7 cells were treated with a series of concentrations of H_2_O_2_. The CCK-8 kit was used to determine cell viability. As shown in Figure 1B, different concentrations of H_2_O_2_ caused different degrees of oxidative stress in the macrophages, resulting in a decrease in the survival rate of the RAW 264.7 cells. When the concentration of H_2_O_2_ was 1000 µM, the viability of the RAW 264.7 cells was only 16.26 ± 1.47%. This shows that H_2_O_2_ causes oxidative damage to RAW 264.7 cells, leading to cell damage and death. The median inhibitory concentration (IC_50_) of H_2_O_2_ for RAW 264.7 cells was 391.4 µM. In order to facilitate subsequent experiments, a model concentration of 400 µM was chosen to establish an oxidative stress model for RAW 264.7 cells.

#### 2.1.3. Cell Viability Tests

In order to determine whether *A. senticosus* flavonoids could protect RAW 264.7 cells from oxidative stress, RAW 264.7 cells were pretreated with Vitamin C (Vc) (100 mg/L) or *A. senticosus* flavonoids (30, 60, or 120 mg/L) for 24 h. Then, after H_2_O_2_ treatment for 2 h, cell viability was directly detected by the CCK-8 kit. As shown in Figure 1C, the cell viability of the model group was significantly lower than that of the blank group (*p* < 0.01), indicating that a RAW 264.7-cell oxidative stress model had been established. Compared with that for those in the model group, the survival of the RAW 264.7 cells in the Vc group and *A. senticosus* group (30, 60, and 120 mg/L) increased (*p* < 0.01). This shows that *A. senticosus* flavonoids and Vc have similar effects, and both can effectively inhibit the oxidative stress induced by H_2_O_2_ in cells and improve cell survival number. Therefore, *A. senticosus* flavonoids have a strong protective effect against oxidative stress induced by the H_2_O_2_-induced RAW 264.7 mouse macrophages, which provides a reference for subsequent experiments to explore the molecular antioxidant mechanisms of *A. senticosus* flavonoids.

#### 2.1.4. Intracellular ROS Level Detection

Excessive ROS can cause oxidative stress in cells. After RAW 264.7 cells were stimulated by H_2_O_2_ for 2 h, a DCFH-DA probe was added, followed by incubation for 20 min such that the probe could effectively enter the cell and bind to ROS. A fluorescence microscope was used to observe the green fluorescence in the blue channel. As shown in Figure 2A, the green fluorescence in the H_2_O_2_ group and the low-dose group increased significantly, and the green fluorescence in the medium-dose group and the high-dose group had a significant downward trend compared with that in the H_2_O_2_ group. Flow cytometry was used to measure the relative content of intracellular ROS (compared with the blank group). The results are shown in Figure 2B,C. As the *A. senticosus* flavonoid concentration increased, the intracellular ROS level gradually decreased. The ROS level in the high-dose group was significantly lower than that in the H_2_O_2_ group (*p* < 0.05). The results show that the flavonoids can prevent the accumulation of excess ROS in RAW264.7 macrophages. When a cell is in an oxidizing environment for a long time, the redox balance of the cell is disrupted, resulting in cell damage and the loss of its original function. *A. senticosus* flavonoids can effectively improve the ability of RAW 264.7 cells by eliminate ROS, showing a dose-dependent manner, thereby enhancing the antioxidant capacity of cells.

#### 2.1.5. Determination of Contents of MDA, CAT, SOD, and GPx

By evaluating the lipid peroxidation and antioxidant enzyme activities in *A. senticosus* flavonoid-treated cells under oxidative stress, the antioxidant effect of *A. senticosus* flavonoids can be further explored. As shown in Figure 3A, when the cells were stimulated by H_2_O_2_ for 2 h, the MDA content in the model group was significantly higher than that in the blank group. The test results for the *A. senticosus* group and the Vc group were similar, showing that the flavonoids could inhibit the production of MDA under oxidative stress, and it was significantly lower than that for the model group (*p* < 0.01). In Figure 3B–D, it can be observed that the activities of the three main antioxidant enzymes—CAT, SOD, and GPx—in the model group were all reduced compared with those in the blank group. As shown in Figure 3B, compared with that of the model group, the CAT activity of the *A. senticosus* group was significantly increased (*p* < 0.01). In the high-dose group, the CAT activity was higher than that in the Vc group. As shown in Figure 3C, the SOD activity was positively correlated with the *A. senticosus* flavonoid concentration, and it was significantly higher than that in the model group *(p* < 0.01). As shown in Figure 3D, the GPx activity of the medium-dose group was slightly higher than that of the model group (*p* < 0.05), while the GPx activity of the high-dose group was higher (*p* < 0.01). The results show that antioxidant enzymes can decrease the damage of H_2_O_2_ in cells, but excessive H_2_O_2_ leads to the excessive consumption of cell antioxidant enzymes, leading to lipid peroxidation and, ultimately, cell death. *A. senticosus* flavonoids can exhibit the same antioxidant effect as Vc and can increase the activity of antioxidant enzymes in RAW 264.7 cells, thereby inhibiting the production of MDA and promoting the maintenance of the normal physiological activities of cells.

#### 2.1.6. Western Blot Assay

In order to further explore the effects of *A. senticosus* flavonoids on protein expression in the protection against oxidative stress, RAW 264.7 cells were treated as above, and then, the cells were lysed on ice. Western blot assay was used to detect the changes in Nrf2 protein content in the nucleus and cytoplasm and the expression of the Keap1 and HO-1 proteins. The result is shown in Figure 4A. The analysis of the gray values of the bands is shown in Figure 4B–E, and it can be observed that the Nrf2 protein content in the nucleus of the model group increased compared with that in the blank group. This shows that, under H_2_O_2_ stimulation, cells can transfer the Nrf2 protein to the nucleus to activate antioxidant activity. The Nrf2 protein contents in the nucleus for the Vc group and the *A. senticosus* group were significantly increased (*p* < 0.01), and the expression of the Nrf2 protein in the cytoplasm showed a downward trend compared with that for the model group (*p* < 0.01). This means that *A. senticosus* flavonoids can promote the transfer of the Nrf2 protein from the cytoplasm to the nucleus in RAW 264.7 cells, thereby activating the antioxidant activity of the cells. As shown in Figure 4D, the expression level of Keap1 protein in the model group was lower than that in the blank group. The Keap1 protein in the cytoplasm was degraded, and the bound Nrf2 protein was released, allowing it to enter the nucleus and activate the antioxidant pathway. The expression of the Keap1 protein in the medium-dose group was lower than that in the model group (*p* < 0.05), and the expression of the Keap1 protein in the high-dose group was also lower than that in the model group (*p* < 0.01). This shows that *A. senticosus* flavonoids can effectively inhibit the expression of the Keap1 protein in macrophages, thereby increasing the level of Nrf2 in the nucleus. As shown in Figure 4E, the expression of the HO-1 protein in the *A. senticosus* group showed an upward trend and was significantly higher than that in the model dose group *(p* < 0.01). This means that *A. senticosus* flavonoids can promote the expression of the HO-1 protein. In summary, *A. senticosus* flavonoids can increase the antioxidant activity in RAW 264.7 cells because they can regulate the expression of antioxidant proteins in the cell. By inhibiting the expression of the Keap1 protein, *A. senticosus* flavonoids can promote the transfer of Nrf2 to the nucleus, thereby increasing the expression of the HO-1 protein and significantly enhancing antioxidant activity.

### 2.2. In Vivo Experiment

#### 2.2.1. The Protective Effects of *A. senticosus* Flavonoids on DSS-Induced Colitis

Weight loss, loose stools, bleeding, and diarrhea are all characteristic symptoms that can be directly observed in DSS-treated mice. It can be seen from Figure 5A that the weights of the mice in the blank group increased throughout the process, while the weights of the mice freely drinking 2.5% DSS showed a downward trend from the fifth day. *A. senticosus* flavonoids can act similarly to mesalazine, a drug commonly used for the treatment of UC, and can effectively improve weight loss in mice. The greater the concentration of *A. senticosus* flavonoids, the slower the rate of weight loss in the mice. Figure 5B shows a photo of the colon of each group. The shortened colon length is also an important indicator that indirectly reflects the severity of ulcerative colitis in mice. The colons of the mice in the DSS group became shorter in length, and diffuse bleeding points appeared in some intestinal segments. The colons in the *A. senticosus* groups were longer than those in the DSS group. Among them, the medium-dose group and the high-dose group showed effective maintenance of the length of the colon without diffuse bleeding. As shown in Figure 5C, compared with those in the DSS group, the colon lengths of the *A. senticosus* groups and the mesalazine group significantly increased (*p* < 0.01), and the effect in the high-dose group was better than that in the mesalazine group. The DAI score is another important indicator for comprehensively evaluating the degree of ulcerative colitis in mice. The mice in each group were scored according to the DAI scoring rules, and the results are shown in Figure 5D. The mice in the blank group were in a good mental state, with normal stools, and their body weights continued to increase. The DAI score was close to 0 throughout the process. From the fourth day onward, except for those in the blank group, the DAI scores of the mice in the groups gradually increased. The mice in the DSS group were depressed, their weights decreased significantly, their stools were loose and shapeless, and severe cases showed symptoms of diarrhea and blood in the stools. Although the mesalazine group and the *A. senticosus* group had the same conditions, the weight loss and fecal conditions were relieved to a great extent compared with those in the DSS group. This manifested as loose stools, hidden bleeding, no diarrhea, and blood in the stool. In summary, *A. senticosus* flavonoids have a certain protective effect on DSS-induced UC in mice.

#### 2.2.2. Histopathological Observation of Colons

UC shows many histological features, such as colonic mucosal erosion, bleeding, shrinkage, the disappearance of crypts, and many inflammatory cell infiltrations. In order to further confirm the protective effect of *A. senticosus* flavonoids on DSS-induced colitis in mice, colon tissue sections from each group of mice were observed under a microscope, and pathological changes were analyzed. As shown in Figure 6A, the mice in the blank group had complete colon tissue structures. The goblet cells on the mucosal surface were arranged neatly, and the villi and crypts of the small intestine were clearly visible. The number of internal glands was large and orderly, without inflammatory cell infiltration. The colonic mucosa of the mice in the DSS group was seriously injured. Specific manifestations included the shedding of goblet cells, the disappearance of glands, the shortening and adhesion of small intestinal villi, the almost complete degeneration and disappearance of crypts, and the appearance of a large number of inflammatory cell infiltrations. Compared with the DSS group, the *A. senticosus* low-dose group showed a smaller effect in terms of UC improvement, with only a small number of goblet cells and fewer glands. The medium-dose group of *A. senticosus* showed a more obvious improvement effect on the colonic mucosa. Some of the goblet cells were arranged in an orderly manner, with a large number of glands but uneven distribution. The overall structure was less damaged, but the inflammatory cell infiltration was more obvious. The *A. senticosus* high-dose group showed the best improvement effect on UC. The structure of the mouse colon was complete, and the small intestinal villi were longer. The glands were evenly distributed, the goblet cells in the mucosal layer were arranged neatly, the crypts were deeper, and the inflammatory cell infiltration was less. The high dose of *A. senticosus* showed a stronger protective effect on UC than the mesalazine group.

The histopathological scores of the mice in each group are shown in Figure 6B. The scores of the DSS group were significantly higher, indicating that the colitis model for the DSS group was successfully established. Compared with those of the DSS group, the histopathological scores of the mesalazine group, the *A. senticosus* medium-dose group, and the *A. senticosus* high-dose group were all significantly reduced (*p* < 0.01). The results show that *A. senticosus* flavonoids have an effect on DSS-induced colitis similar to that of mesalazine and can effectively protect the colon from the damage caused by DSS.

#### 2.2.3. The Contents of MDA, CAT, SOD, and GPx in Colon Tissue

In order to explore whether *A. senticosus* flavonoids could inhibit the DSS-induced UC by increasing the body’s antioxidant activity, the experiment involved extracting the protein in the colon tissue of each group, and the content of MDA and the activities of CAT, SOD, and GPx were analyzed. As shown in Figure 7A, the content of MDA in the DSS group was significantly higher than that in the blank group (*p* < 0.01), indicating that there was lipid peroxidation in the colons of the DSS group. The *A. senticosus* groups showed a decrease in the content of MDA in the colon tissue, which was lower than that in the DSS group (*p* < 0.01). As shown in Figure 7B–D, the activities of CAT, SOD, and GPx in the colon tissues of the DSS group all decreased compared with those in the blank group (*p* < 0.01), indicating that the colons of mice in the DSS group had severe oxidative stress. The antioxidant enzyme activity of the mice colon in the mesalazine group was significantly increased and higher than that in the DSS group (*p* < 0.01). *A. senticosus* flavonoids have a similar effect to mesalazine; it is dose-dependent, and the higher the concentration, the better the effects of the CAT, SOD, and GPx activities. Compared with those in the DSS group, the activities of CAT, SOD, and GPx in the high-dose group increased (*p* < 0.01). It is concluded that DSS induces severe oxidative stress in the colon in the mouse colitis model, while *A. senticosus* flavonoids can effectively inhibit colitis by improving the body’s antioxidant capacity.

## 3. Discussion

Studies have found that, when phagocytes in the body receive stimulation from external factors, the body produces a certain amount of reactive oxygen species, which promotes the phagocytic and killing effects of phagocytes [17]. Excessive ROS are easily converted into H_2_O_2_ in the cell, which then diffuses through the cell membrane. The Fenton reaction catalyzed by heme produces highly reactive and toxic hydroxyl free radicals, which ultimately triggers the death of macrophages [20,25,26]. The virulence of some bacteria triggers the death of macrophages by stimulating the production of ROS [27]. Macrophages participate in the host defense system and eliminate pathogens such as microorganisms, and they are sensitive to ROS responses. Therefore, in this study, macrophages were selected to establish an oxidative stress model. The cell experiments proved that *A. senticosus* flavonoids could increase the activity of CAT, SOD, and GPx and reduce the levels of ROS and MDA in cells, while also inhibiting cell lipid peroxidation. They can also activate HO-1 gene activity through inducing the Nrf2 protein to activate the transcription of the HO-1 gene, inhibit cell oxidative stress, and maintain normal cell physiological activity.

Regarding animal experiments, DSS is a large molecule and negatively charged compound that can easily reach the colon and be absorbed following free drinking water administration. The mechanism of the induction of colitis may be the high salt properties leading to an imbalance in the osmotic pressure inside and outside the colon, thereby damaging the colonic epithelial tissue [28]. In this experiment, C57BL/6 mice were allowed to freely drink 2.5% DSS solution for 7 days to successfully establish a mouse colitis model. In the DSS group, the mice showed symptoms of weight loss, diarrhea, blood in the stool, and shortened colon length. The treatment group administered the *A. senticosus* flavonoid extract solution showed that weight loss, diarrhea and shortening of the colon length of the mice, the characteristic symptoms of treatment groups, were significantly reversed and improved than in that of the DSS group. The results of H&E staining were consistent with the results of the DAI evaluation. In the DSS group, the level of inflammatory infiltration was higher, and the villi were shortened and shedding. Compared with the DSS-induced model group, the area of inflammatory cell infiltration and colon ulcers in the *A. senticosus* group was significantly reduced, and the adhesion and shortening of villi were also greatly improved. Many studies have shown that UC can reduce the activity of antioxidant enzymes in colon tissue, thereby causing local lipid peroxidation [29,30]. In this experiment, the activities of CAT, SOD, and GPx in the DSS group mice were significantly reduced, but the MDA content increased. This showed that oxidative stress and lipid peroxidation in colon tissue were intensified and that the antioxidant mechanism was severely damaged. Under the treatment of *A. senticosus* flavonoids, the activity of three antioxidant enzymes in colon tissue was significantly increased, and the content of MDA decreased. Therefore, *A. senticosus* flavonoids can effectively inhibit the colitis induced by DSS, which has important implications for protecting and improving intestinal homeostasis.

## 4. Materials and Methods

### 4.1. Chemicals, Reagents, and Animals

*A. senticosus* was purchased from Liaocheng Limin Pharmacy (Liaocheng, China), ground into a powder with a grinder (H8422, Hebei Huicai Technology Co. Ltd., Hebei, China), and then sieved through a 60-mesh sieve and stored at room temperature. Fetal bovine serum was obtained from Wolcavi Co., Ltd. (Beijing, China). The 1640 medium was purchased from Hyclone Co., Ltd. (Logan, UT, USA). A reactive oxygen species detection kit, BCA protein concentration detection kit, lipid oxidation detection kit, glutathione peroxidase detection kit, catalase detection kit, superoxide dismutase activity detection kit, SDS-PAGE gel-preparation kit, RIPA Lysis Solution, and 4% paraformaldehyde were all purchased from Biyuntian Biotechnology Co., Ltd. (Shanghai, China). Nrf2 Monoclonal Antibody, Keap1 Monoclonal Antibody, HO-1 Monoclonal Antibody, Beta Actin Monoclonal Antibody, Lamin B1 Monoclonal Antibody, and Goat anti-mouse IgG (H + L) were prepared from Proteintech Biotechnology Co., Ltd. (Wuhan, China). DSS was obtained from Bomei Biotechnology Co., Ltd. (Anhui, China). Mesalazine enteric tablets were purchased from Tianhong Pharmaceutical Co., Ltd. (Heilongjiang, China).

RAW 264.7 cells were purchased from the Stem Cell Bank of the Chinese Academy of Sciences (Beijing, China).

Animals: 60 SPF male C57BL/6 mice, 8–10 weeks old, weighing 22 ± 2 g, were purchased from Jinan Pengyue Experimental Animal Breeding Co., Ltd. (Shandong, China). The mice were raised in an animal room of free from specific pathogens. The temperature was controlled at 24 ± 2 °C, the relative humidity was 60 ± 5%, and the light–dark cycle was 12 h. All the experimental mice had free access to water and food, and all survived in the DSS feeding process.

### 4.2. Animals’ Care

The mice study was carried out in strict accordance with the recommendations in the Guide for the Care and Use of Laboratory Animals. According to the guidelines, ether was used to euthanize the mice. The protocols for animal studies were approved by the Committee on the Ethics of Animal Experiments of Liaocheng University and Use Committee (LCU-2020-0126).

### 4.3. In Vitro Experiment

#### 4.3.1. Preparation of *A. senticosus* Flavonoid Extract

A certain amount of *A. senticosus* powder was placed in an Erlenmeyer flask, and a 55% ethanol aqueous solution was added at a solid–liquid ratio of 1:45 (g/mL). Under the conditions of an ultrasonic power of 300 W and extraction temperature of 72 °C, the sample was extracted for 73 min to obtain a crude extract of *A. senticosus* flavonoid. After adsorption by AB-8 macroporous resin and desorption using 60% ethanol solution, the *A. senticosus* flavonoid solution was purified and stored at −20 °C for later use.

#### 4.3.2. Determination of Total Flavonoids Content

The content of total flavonoids was determined using the NaNO_2_-Al(NO_3_)_3_-NaOH colorimetric method described by Feng et al. [31] with slight modifications. The rutin was used as the reference. The freeze-dried *A. senticosus* extract was dissolved in water to prepare a stock liquid (the concentration of total flavonoids: 500 mg/L, with a purity of 47.65 ± 0.032%). The stock liquid was diluted into a concentration gradient diluent for subsequent experiments.

#### 4.3.3. Cell Culture

RAW 264.7 cells were cultured in a 1640 medium containing 10% fetal bovine serum and in a humid incubator at 37 °C with 5% CO_2_. When the cell density reached 80~90%, cell passages was carried out.

#### 4.3.4. Cytotoxicity Test

According to the literature [32], the CCK-8 kit was used for the detection of drug toxicity. The cell density was diluted to 5 × 10^4^/mL, and 200 µL of cell suspension was seeded to each well of a 96-well plate. When the cells grew to 80~90% confluence, the supernatant was discarded, and the cells were washed three times with PBS. Volumes of 200 µL of 1640 medium containing different concentrations of *A. senticosus* flavonoids (final concentrations of 15, 30, 60, 90, 120, and 150 mg/L) were added to the cells, and the supernatant was discarded after 24 h of treatment. Some 1640 medium containing 10% CCK-8 solution was added to each well, and the cells were then incubated at 37 °C for 30 min. The absorbance at 450 nm was measured with a full-wavelength microplate reader, and then the cell viability was calculated. Serum-free medium was used in the blank group, and 6 replicate wells were established in each group.

#### 4.3.5. Establishment of H_2_O_2_-Induced Oxidative Damage Model in RAW 264.7 Cells

Cells were plated in a 96-well plate at a density of 5 × 10^4^ cells, with 200 µL of medium in each well. When the cells grew to 80~90% confluence, the supernatant was discarded, and the cells were washed with PBS. Volumes of 200 µL of serum-free 1640 medium containing different concentrations of H_2_O_2_ were added, and the cells were incubated for 2 h. The CCK-8 kit was used to determine the cell survival rate. For the blank control group, serum-free 1640 medium without H_2_O_2_ was used, and 6 replicate wells were established in each group.

#### 4.3.6. Cell Viability Tests

The cells were divided into a blank group, H_2_O_2_ group, Vc group (100 mg/L), *A. senticosus* low-dose (30 mg/L) group, *A. senticosus* medium-dose (60 mg/L) group, and *A. senticosus* high-dose (120 mg/L) group. When the cells grew to 80~90% confluence, the medium was removed, and the cells were washed with PBS. The *A. senticosus* group was treated with the corresponding amount of *A. senticosus* flavonoids for 24 h, and the blank group and H_2_O_2_ group were treated with serum-free 1640 medium instead of *A. senticosus* flavonoids. Then, the medium was removed, and the cells were washed with PBS again. Then, serum-free 1640 medium containing 400 µM H_2_O_2_ was added to it for 2 h. In the blank group, serum-free 1640 medium was used instead of H_2_O_2_. The CCK-8 kit was used to calculate the cell viability [33].

#### 4.3.7. Detection of Intracellular ROS Level

RAW 264.7 cells were evenly seeded in a 6-well plate and cultured according to Section 4.3.6., and this was repeated 3 times for each group. After the cells had been washed with 1640 medium, 500 µL of 1640 medium containing 10 µM DCFH-DA was added to them. After the treatment, the cells were incubated in a cell incubator in the dark for 20 min. The supernatant was discarded, and the cells were washed three times with 1640 medium to fully remove the probes that did not enter the cells. A fluorescence microscope was used to observe the distribution of ROS in the cells. After processing according to the above method, the cells were collected, and the fluorescence intensity of intracellular ROS was analyzed using the NovoCyte flow cytometer.

#### 4.3.8. Detection of the Contents of MDA, CAT, SOD, and GPx

The cells were seeded in a 6-well plate at a cell density of 2 × 10^5^/mL. When the cell density reached 80–90%, the supernatant was discarded, and the cells were washed with PBS. The cells were cultured according to the method in Section 4.3.6. The cells were lysed according to the kit instructions, and the supernatant was collected. The protein concentration was detected by the BCA method, and the contents of MDA, CAT, SOD, and GPx in the cells were detected in strict accordance with the instructions of the kit. Finally, the relative content was calculated based on the protein concentration of the sample.

#### 4.3.9. Western Blot Detection

RAW 264.7 cells were treated with *A. senticosus* flavonoids for 24 h and then stimulated with 400 µM H_2_O_2_ for 2 h. After the treatment, the cells were collected, and the nuclear protein and plasma protein were extracted using the nuclear protein and cytoplasmic protein extraction kit. The protein concentration was measured using the BCA method. The SDS-PAGE protein loading buffer was added to the protein sample, and the protein was denatured by heating at 100 °C for 10 min using a heating block. An amount of 20 µg of protein was sampled for 10% SDS-PAGE. Referring to the method in the literature [34], after separation, the protein was transferred to the PVDF membrane by the wet transfer method, and the membrane was blocked using 5% skim milk powder for 3 h. The diluted primary antibody was incubated with the membrane overnight at 4 °C (dilution ratios: β-actin, 1:10,000; Lamin B1, 1:10,000; Nrf2, 3:10,000; Keap1, 2:10,000; HO-1, 2:10,000). The next day, the membrane was washed 3 times for 100 min using TBST. After the secondary antibody was added, it was incubated at room temperature for 1.5 h (source: goat; dilution ratio: 1:10,000). Then, it was washed 3 times for 15 min with TBST. The treated PVDF membrane was visualized by the ECL chemiluminescence method, and the relative protein expression was analyzed.

### 4.4. In Vivo Experiment

#### 4.4.1. Establishment of DSS-Induced Colitis Model

With reference to research methods such as of Chaudhary G. [35] and Marín [36], accurately weighed 2.5 g of DSS powder was added to 100 mL of sterile water and fully stirred until completely dissolved, and a 2.5% DSS aqueous solution was obtained. Mice drank the solution freely for 7 days to establish a mouse model of colitis.

#### 4.4.2. Animal Study

Sixty SPF male C57BL/6 mice were randomly divided into 6 groups (n = 10): a blank group, DSS group, mesalazine group (25 g/L), *A. senticosus* low-dose group (50 g/L), *A. senticosus* medium-dose group (100 g/L), and *A. senticosus* high-dose group (200 g/L). The test period was 10 days. The mice in each group were intragastrically administered a dose of 10 mg per kilogram of body weight at 16:00 every afternoon. The mice in the blank group and the DSS group were given normal saline, the mice of the mesalazine group were given mesalazine solution, and the mice in the *A. senticosus* groups were given different doses of *A. senticosus* flavonoids. During the 1st to 3rd day of the experiment, all the groups drank pure drinking water. On the 4th to 10th days, the mice in the blank group continued to drink pure drinking water, but the remaining groups drank 2.5% DSS aqueous solution. The experimental process is shown in Figure 8. During the process, the mice were weighed daily, fecal conditions were observed, and fecal occult blood tests were performed. On the 11th day, the mice were sacrificed by cervical dislocation after ether anesthesia. At the same time, the intestinal tissues from the cecum to the anus were dissected, and the length was measured. According to Table 1, the disease activity index (DAI) of each mouse was determined.

#### 4.4.3. Observation of Colonic Histopathology

Fresh colon tissue was cut into 1 cm sections and fixed with 4% paraformaldehyde solution for 24 h. The pathological tissue sections were made and stained with Hematoxylin–Eosin Y (H&E). The colon sections of each group were observed under a microscope, and the pathological changes were analyzed. According to the method of Siegmund et al. [37], the histopathological scores were determined as shown in Table 2.

#### 4.4.4. Measurement of the Contents of MDA, CAT, SOD, and GPx

Under ice-bath conditions, fresh colon tissue was minced and homogenized with RIPA lysis buffer. After the homogenate was centrifuged at 12,000 rpm at 4 °C for 20 min, the supernatant was stored at 4 °C for later use. The BCA method was used to measure the protein concentration in the supernatant. Strictly following the instructions for the kit, chemical chromatography was used to measure the content of MDA and the activities of CAT, SOD, and GPx in the colon tissue.

### 4.5. Data Analysis

Each set of tests was performed in triplicate, and the test results are expressed as the means ± standard deviations. GraphPad Prism 8.0.1 software (GraphPad Corporation, San Diego, CA, USA) was used for mapping, ImageJ software was used to analyze the Western blot grayscale images, and SPSS 25 was used for multiple-comparison with Tukey’s post hoc test and the significance analysis of the data.

## 5. Conclusions

Taken together, the current results demonstrate that *A. senticosus* flavonoids produce significant antioxidant activity in H_2_O_2_-induced RAW 264.7 cells and DSS-induced colitis in mice. The findings from our study prove that *A. senticosus* flavonoids can not only protect macrophages under oxidative stress by increasing the activity of antioxidant enzymes in the cell and activating the Nrf2/Keap1/HO-1 signaling pathway but also inhibit colitis by improving antioxidant activity in mice and maintaining the normal physiological function of the intestinal tract. These findings provide evidence that the total flavonoids of *A. senticosus* can serve as potent compounds for use in the treatment of diseases related to oxidative stress.

## Figures and Tables

**Figure 1 molecules-27-02872-f001:**
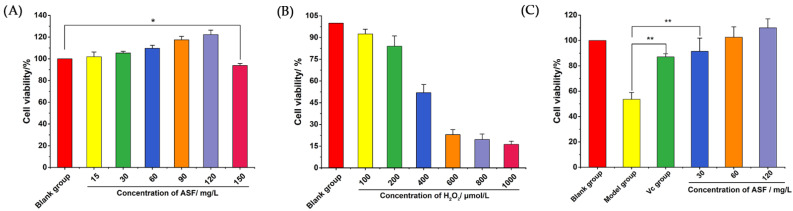
Results of toxicity test. (**A**) Effects of total flavonoids of *A. senticosus* on the viability of RAW 264.7 cells. (**B**) The effects of H_2_O_2_ on proliferation of RAW 264.7 cells. (**C**) The protective effects of total flavonoids on oxidative damage of RAW 264.7 cells. Note: For the comparison between the two groups, * indicates significant difference at *p* < 0.05. ** indicates extremely significant difference at *p* < 0.01.

**Figure 2 molecules-27-02872-f002:**
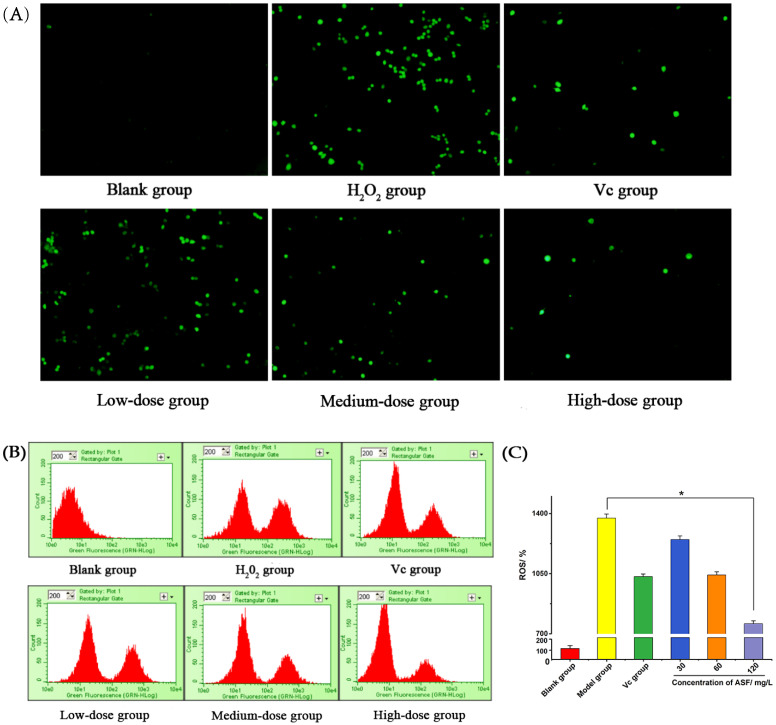
The detection of reactive oxygen species in RAW 264.7. (**A**) Representative image of DCFH-DA immunofluorescence staining. (**B**) Images of the relative content of ROS in cells measured by flow cytometry. (**C**) Quantitative analysis of Western blot data for DCFH-DA markers in RAW 264.7 cells treated with H_2_O_2_ for 2 h. Note: For the comparison between the two groups, * indicates significant difference at *p* < 0.05.

**Figure 3 molecules-27-02872-f003:**
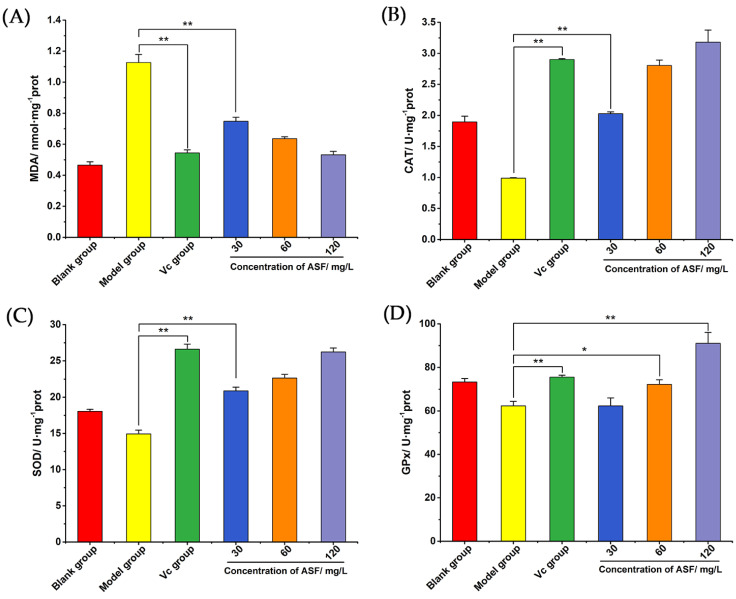
The effects of total flavonoids on MDA and antioxidant enzymes in RAW 264.7 cells. (**A**) The effect of *A. senticosus* flavonoids on the cellular concentration of MDA. (**B**) The effect of *A. senticosus* flavonoids on the activity of CAT. (**C**) The effect of *A. senticosus* flavonoids on the activity of SOD. (**D**) The effect of *A. senticosus* flavonoids on the activity of GPx. Note: For the comparison between the two groups, * indicates significant difference at *p* < 0.05. ** indicates extremely significant difference at *p* < 0.01.

**Figure 4 molecules-27-02872-f004:**
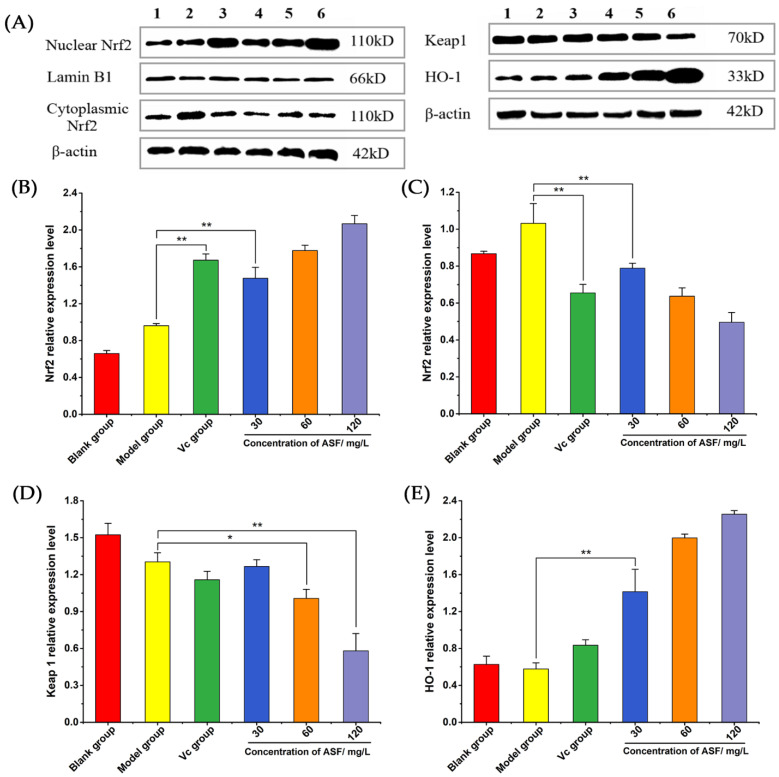
The expression level of Nrf2, Keap1, and HO-1 protein. (**A**) The bands of protein expression: 1, blank group; 2, model group; 3, Vc group; 4, low-dose group (30 mg/L); 5, medium-dose group (60 mg/L); 6, high-dose group (120 mg/L). (**B**) The effect of *A. senticosus* flavonoids on the relative level of Nrf2 nucleoprotein. (**C**) The effect of *A. senticosus* flavonoids on the relative level of Nrf2 cytoplasmic protein. (**D**) The influence of *A. senticosus* flavonoids on the relative expression of Keap1 protein. (**E**) The effect of *A. senticosus* flavonoids on the relative expression of HO-1 protein. Note: For the comparison between the two groups, * indicates significant difference at *p* < 0.05. ** indicates extremely significant difference at *p* < 0.01.

**Figure 5 molecules-27-02872-f005:**
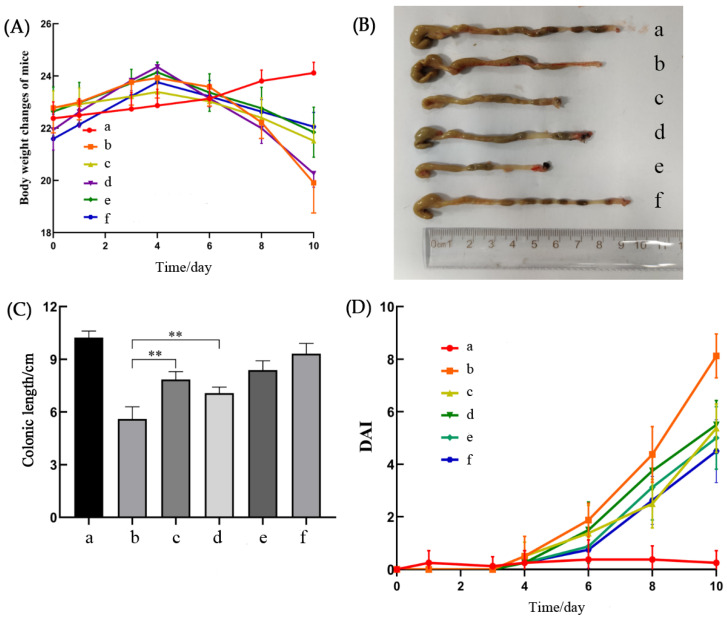
The protective effect of total flavonoids of *A. senticosus* on DSS-induced UC. (**A**) Measurement results for changes in body weights of mice in each group. (**B**) Picture comparing the lengths of the colons of each group of mice. (**C**) Results of measuring the colon length for each group of mice. (**D**) During the experiment, the trend of the DAI score of each group of mice. Legend description: a, blank group; b, DSS group; c, mesalazine group; d, *A. senticosus* low-dose group; e, *A. senticosus* medium-dose group; f, *A. senticosus* high-dose group. Note: For the comparison between the two groups, ** indicates extremely significant difference at *p* < 0.01.

**Figure 6 molecules-27-02872-f006:**
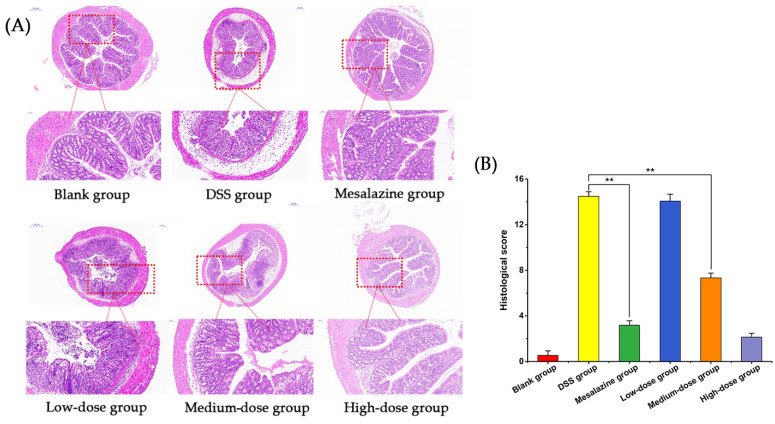
Effects of *A. senticosus* on colonic pathological changes in mice with DSS-induced colitis. (**A**) Mouse colonic tissue sections (magnification: ×40 and ×200). (**B**) Histopathological scoring of mouse colons. Note: For the comparison between the two groups, ** indicates extremely significant difference at *p* < 0.01.

**Figure 7 molecules-27-02872-f007:**
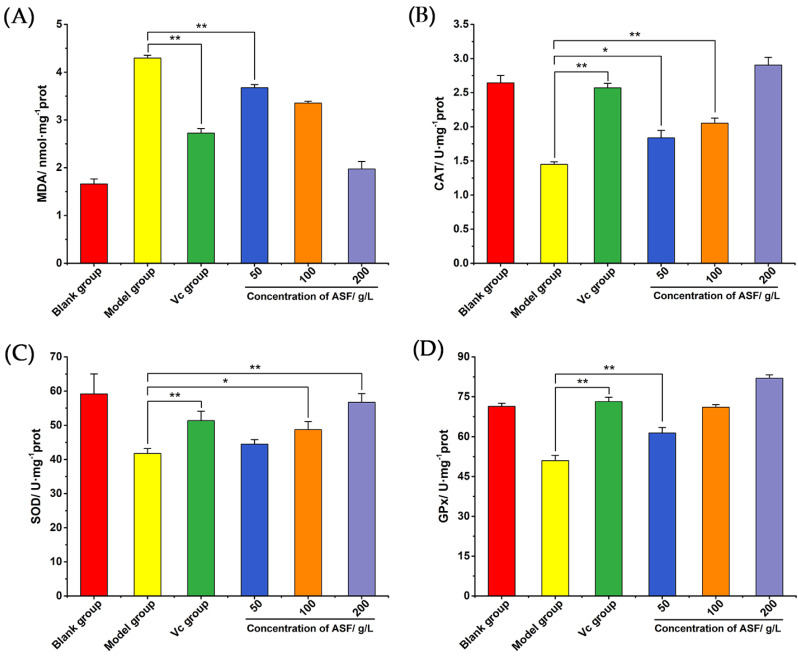
The effects of total flavonoids on MDA and antioxidant enzymes in the colon. (**A**) The effect of *A. senticosus* flavonoids on the content of MDA in colon tissue. (**B**) The effect of *A. senticosus* flavonoids on the activity of CAT. (**C**) The effect of *A. senticosus* flavonoids on the activity of SOD. (**D**) The effect of *A. senticosus* flavonoids on the activity of GPx. Note: For the comparison between the two groups, * indicates significant difference at *p* < 0.05. ** indicates extremely significant difference at *p* < 0.01.

**Figure 8 molecules-27-02872-f008:**
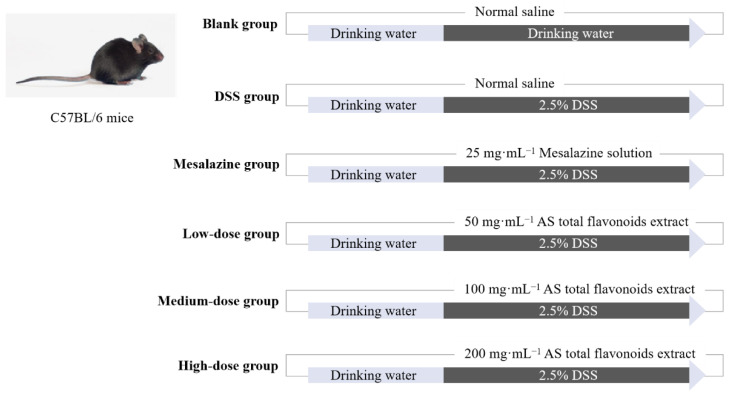
The procedure for testing the inhibition of DSS-induced UC by total flavonoids of *Acanthopanax*.

**Table 1 molecules-27-02872-t001:** The DAI scores.

Score	Weight Loss [%]	Stool Consistency	Blood Stool
0	0	Normal	Normal
1	1~5	Slightly loose	Normal
2	5~10	Loose	Occult bleeding
3	10~15	Sticky	Occult bleeding
4	15~	Watery diarrhea	Overt bleeding

**Table 2 molecules-27-02872-t002:** The histopathological scores.

Score	0	1	2	3	4
Percent tissue damage	None	<25%	<50%	<75%	<100%
Extent of tissue damage	None	Mucosal layer	Submucosa	Muscle layer	Outer membrane
Degree of inflammation	None	Slight	Moderate	Severe	
Extent of crypt damage	None	Damage 1/3	Damage 2/3	Completely disappeared	

## Data Availability

Not applicable.

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
