# Peer review of "Antioxidant Activity of Acanthopanax senticosus Flavonoids in H2O2-Induced RAW 264.7 Cells and DSS-Induced Colitis in Mice"

_molecules, 2022, doi:10.3390/molecules27092872_

Round 1

Reviewer 1 Report

The manuscript is well written and is scientifically sound. The work is significant and would be of high interest to the readers. The manuscript could use following correction, details are highlighted in the attached pdf.

  1. Introduction could use clear, short and active sentences
  2. A thesis statement is must
  3. Citations should follow journal guidelines
  4. Check to include active sentences throughout the manuscript.
  5. Possibly include ethical committee details
  6. References should use uniform format.

Author Response

Dear reviewer,

Thank you very much to review my manuscript (molecules-1699332), and give me some suggestions. These comments are valuable and very helpful. We have read through comments carefully and have made corrections. Based on the instructions you provided, we uploaded the field of the revised manuscript. Revisions in the text are shown using red highlight for additions, and strike through font for deletions. The responses to your comments are marked in red and presented following in this word.

We would love to thank you for allowing us to resubmit a revised copy of the manuscript and we highly appreciate your time and consideration.

Yours sincerely,

                                                            Jianqing Su

General Comments:

Comment 1: Introduction could use clear, short and active sentences.

Response 1: Thank you for your suggestion, I have revised the expression style of sentence in the introduction.

Comment 2: A thesis statement is must.

Response 2: I have added the thesis statement in paper.

Comment 3: Citations should follow journal guidelines

Response 3: I have revised the citations by journal guidelines.

Comment 4: Check to include active sentences throughout the manuscript.

Response 4: Thanks for your suggestion, I have carefully revised the sentences throughout the manuscript.

Comment 5: Possibly include ethical committee details.

Response 5: I have supplemented ethical committee details. Line 395

Comment 6: References should use uniform format.

Response 6: I have reviewed the references format by journal guidelines.

And complete the modification according to the comments in the attachment.

Reviewer 2 Report

I would like to inform you that,

The manuscript entitled "Antioxidant Activity of Acanthopanax senticosus flavonoids in  H2O2-Induced RAW 264.7 Cells and DSS-Induced Colitis in 3 Mice"

Carful inspection with our respects for the efforts which were done before, the manuscript does not show any new points of innovation or any kind of fascinating research.

As, iam strongly believe especially the figures (4-6) of Western blotting, should be please submit the whole length blots for each protein target as supplementary figures. In addition to, all flavonoids are known for their cytoprotective actions.

So, i reject to inform you that the manuscript is suffer from a clear data with the lacked of novelty. I am afraid of the manuscript should be refused and rejected and not suitable to be published in the valuable journal like molecules.

Author Response

Dear reviewer,

Thank you very much to review my manuscript (molecules-1699332), and give me some suggestions. These comments are valuable and very helpful. We have read through comments carefully and have made corrections. Based on the instructions you provided, we uploaded the field of the revised manuscript. Revisions in the text are shown using red highlight for additions, and strike through font for deletions. The responses to your comments are marked in red and presented following in this word.

We would love to thank you for allowing us to resubmit a revised copy of the manuscript and we highly appreciate your time and consideration.

Yours sincerely,

                                                            Jianqing Su

General Comments:

Comment 1: Careful inspection with our respects for the efforts which were done before, the manuscript does not show any new points of innovation or any kind of fascinating research.

Response 1: Thanks for your comments, we had found the traditional Chinese medicine, Acanthopanax senticosus, which has strong antioxidant effects in clinical application. Through laboratory research, the flavonoids are main activity component associated with antioxidant activity in A. senticosus, but its mechanism was very little studied. We thought to study the antioxidant activity of flavonoids of A. senticosus in vitro and in vivo. The result showed that flavonoids of A. senticosus exerted antioxidant activity by Increase the contents of intracellular antioxidant enzymes and activated of Nrf2 related antioxidant signaling pathways.

As Reviewers’ advice, we will do our best to get fund support in order to study definite ingredients by ultra-high performance liquid chromatography mass spectrometry, and explain its mechanism in Intestine metabolism. This is going to our next experimental content.

Comment 2: As, I am strongly believing especially the figures (4-6) of Western blotting, should be please submit the whole length blots for each protein target as supplementary figures. In addition to, all flavonoids are known for their cytoprotective actions.

Response 2: We have provided our test pictures as supplementary material. I want to explain our experiment steps. The treated target protein samples were separated by Sodium dodecyl sulfate–polyacrylamide gel electrophoresis with maker proteins, and then, the part of polyacrylamide gel contained target proteins bands was cut for western blot analysis. The aim of this operation is to decrease using amount of the primary antibodies and secondary antibodies. Because the price of antibodies is very expensive. Therefore, only the part of the gel containing the target protein band is transferred on a polyvinylidene fluoride (PVDF) membrane.

Reviewer 3 Report

Congratulations on submitting a good piece of work!

A few comments:

  • Graphs should be of high resolution
  • Captions should say what ** mean
  • Name the multicomparison test used?
  • Graphical abstract should be provided

Please accept the paper after incorporation of these changes

Author Response

Dear reviewer,

Thank you very much to review my manuscript (molecules-1699332), and give me some suggestions. These comments are valuable and very helpful. We have read through comments carefully and have made corrections. Based on the instructions you provided, we uploaded the field of the revised manuscript. Revisions in the text are shown using red highlight for additions, and strike through font for deletions. The responses to your comments are marked in red and presented following in this word.

We would love to thank you for allowing us to resubmit a revised copy of the manuscript and we highly appreciate your time and consideration.

Yours sincerely,

                                                            Jianqing Su

General Comments:

Comment 1: Graphs should be of high resolution

Response 1: We will re-revise the resolution of the images as requested by the magazine.

Comment 2: Captions should say what ** mean

Response 2: For the comparison between the two groups, **indicates extremely significant difference at p <0.01.

Comment 3: Name the multicompanies test used?

Response 3: The ANOVA with Tukey post hoc test was used in the multicompanies test. Line 531

Comment 4: Graphical abstract should be provided

Response 4: The Graphical abstract has been provided in paper. Line 29

Round 2

Reviewer 2 Report

The Manuscript is accepted